# Physical Length and Weight Reduction of Humanoid In-Robot Network with Zonal Architecture

**DOI:** 10.3390/s23052627

**Published:** 2023-02-27

**Authors:** Chengyu Cui, Chulsun Park, Sungkwon Park

**Affiliations:** Department of Electronic and Computer Engineering, Hanyang University, Seoul 04763, Republic of Korea

**Keywords:** robot sensor network, humanoid, zonal architecture, domain architecture, in-robot network (IRN), in-vehicle network (IVN), zonal IRN architecture (ZIRA), domain IRN architecture (DIRA), wiring harness, wiring length, wiring weight

## Abstract

Recently, with the continuous increase in the number of sensors, motors, actuators, radars, data processors and other components carried by humanoid robots, the integration of electronic components within a humanoid is also facing new challenges. Therefore, we focus on the development of sensor networks suitable for humanoid robots to designing an in-robot network (IRN) that can support a large sensor network for reliable data exchange. It was shown that the domain based in-vehicle network (IVN) architectures (DIA) used in the traditional and electric vehicles is gradually moving towards zonal IVN architectures (ZIA). Compared with DIA, ZIA for vehicles is known to provide better network scalability, maintenance convenience, shorter harness length, lighter harness weight, lower data transmission delay, and other several advantages. This paper introduces the structural differences between ZIRA and the domain based IRN architecture (DIRA) for humanoids. Additionally, it compares the differences in the length and weight of wiring harnesses of the two architectures. The results show that as the number of electrical components including sensors increases, ZIRA reduces at least 16% compared to DIRA, the wiring harness length, weight, and its cost.

## 1. Introduction

In the robotics research industry, different from general industrial robots, humanoids manufactured for the purpose of realizing human appearance and behavior have a high demand for the integration of electronics and mechanical construction. Compared with industrial robots, humanoid robots need to have a higher awareness of the surrounding environment, a larger range and accuracy of motion, and more intelligent interaction capabilities [1,2,3,4,5,6,7]. In order to realize these requirements, along with the development of the robotics industry, the focus of research has gradually shifted from mechanical construction to electrical and electronic design. Atlas, a humanoid robot with agile movement ability, is powered by an on-board computer to process the environmental data collected from light detection and ranging (LiDAR), stereo camera, and laser range finder, and then transmit control data to the hydraulic drives all over the robot’s body [5,6]. The interactive robot Ameca is equipped with 27 motors on the head to achieve rich facial expression close to human beings [6,7]. Therefore, during the development of humanoid robots, the number and performance of core components such as various types of sensors, drivers, motors, and processors are constantly increasing and improving, respectively. At the same time, to realize the coordinated operation of these components, the research on the internal communication network of the robot also known as in-robot network (IRN) is gaining more attention [8,9,10].

In previous research, the idea of the IRN was proposed, and the network was divided according to the domain structure according to the differentiation of sensors, environmental perception components, processors, and controllers [3,11,12]. This idea is similar to the division of the five domains in the in-vehicle network (IVN) electrical/electronic (E/E) architecture [13,14,15]. In the traditional domain based IVN architecture, the network is usually divided into five different domains: connectivity, driver replacement, powertrain and vehicle dynamics, body and comfort, and in-vehicle experience [16]. This kind of network focuses on the aggregation of functionally related Electronic Control Units (ECUs) under the domain controller or gateway, and it is easier to achieve the integration of similar data. However, the same type of network components will be distributed in various locations of the humanoid robot according to the usage requirements. Therefore, to integrate similar components, the relevant network wiring will extend from the ECU to every corner of the humanoid robot body. The biggest disadvantage of this structure is that it will increase the wiring complexity, weight, and cost. At the same time, it will also increase the difficulty of assembly, modularization, and the plug-and-play function of some parts during upgrade and maintenance.

In the past two years, a novel automotive network architecture design called the zonal architecture has been proposed. Different from the domain architecture, the zonal architecture reduces the wiring and weight of the network at the cost of increasing software complexity. The benefits of this architecture will increase over time as next-generation vehicles require more network elements to support ever-increasing data processing volumes. Because humanoid robots have higher precision requirements for data processing and dynamic control, the integration of the network is much higher than that of vehicles. The ability of the zonal architecture to simplify network hardware has a positive impact on the integration of humanoid robot networks and the modularization of components.

This paper compares the difference between the IRN domain architecture and the zonal architecture in terms of network structure, hardware quantity, and regional division, and focuses on comparing the differences in the wiring harness of the domain and zonal IRN architectures. Section 2 introduces the development status of humanoid robots in recent years, and the development of domain and zonal architecture in the IVN field and compares difference between them. Section 3 describes the domain and zonal architectures designed for the IRN field. Section 4 describes the calculation results of the wiring harness parameters of the domain and zonal architectures. Finally, Section 5 describes the conclusions drawn based on the comparison results.

## 2. Background and Related Works

This section introduces the background knowledge of our research and related works. Section 2.1 introduces the current research status of robots developed for industrial, social, and movement capabilities in the field of humanoid robot research. Section 2.2 introduces the domain IVN architecture (DIA) currently in use and the newly proposed zonal IVN architecture (ZIA).

### 2.1. Development Status in the Humanoid Robot Field

Humanoid robot research started from the imitation of bipedal walking and expanded to the research and development of artificial intelligence. The characteristics of humanoid robots can be summarized into three categories: interaction, perception, and control [3,4]. Humanoid robots need to have strong human-computer interaction capabilities. At the same time, humanoid robots often face more diverse scenarios than traditional robots, and the environment is more uncertain, requiring environmental awareness and unstructured environment operation capabilities. In addition, the body shape of humanoid robots is closer to that of humans, so volume and weight are limited. At the same time, it needs to realize gait walking, which also puts forward higher requirements for motion control.

The ASIMO humanoid robot developed by Japan’s Honda is 1.3 m high and weighs 50 kg. Through the 6-axis foot area sensor, the gyroscope and acceleration sensor implements the collection of environmental data and attitude balance information, and it uses the drivers of head, arms, hands, hips, legs, and other parts to achieve the mobility of 57 degrees of freedom and has a walking speed of 1.7 mph and a running speed of 4.3 mph [11,17,18].

The T-HR3 humanoid robot developed by Japan’s Toyota allows users to obtain the robot’s perspective and control its behavior through wearable devices. An on-board T-HR3 and reduction gears, motors, master maneuvering system, and torque sensors are connected to each joint of the robot body. These modules transmit the operator’s movements directly to the T-HR3′s 29 body parts and the master maneuvering system’s 16 master control systems [19].

The Atlas robot launched by Boston Dynamics has high mobility, flexibility, and moving speed. Atlas uses a head-mounted LiDAR, two stereo RGB cameras, a TOF depth sensor, and a laser rangefinder to collect environmental data. It uses a multi-plane segmentation algorithm to extract the plane from the point cloud, build the model of the different objects seen by Atlas, and then plans a path based on the model built by itself. Custom valves and hydraulic power units are able to power its 28 hydraulic joints for a high degree of flexibility [5].

The interactive robot Ameca, which was proposed in 2022, was developed by the British technology company Engineered Arts. It weighs 49 kg and is 1.87 m high. The body has 52 modules and supports 51 joint movements. Twenty-seven motors are integrated in the face of Ameca, and thus realizes a realistic facial expression control ability close to human beings. However, although Ameca has a realistic face, it cannot walk, jump, or run [7].

In recent years, the development process in the field of humanoid robots has shown a trend—the number of environmental detection components such as sensors, radars, and photoelectric components integrated in the robot body is gradually increasing. At the same time, to achieve more flexible and faster mobility, the number of drive components used in each joint has gradually increased. However, because the existing humanoid robots are all developed for a single field, there is no humanoid robot that integrates artificial intelligence, environmental detection, language analysis, mobility, and other aspects perfectly. One of the reasons is that developers do not have a suitable network architecture that can support such complex communication.

### 2.2. Comparison of Domain IVN Architecture and Zonal IVN Architecture

#### 2.2.1. Domain IVN Architecture

In today’s industry, the automotive E/E architecture is divided into five areas by function: power domain (powertrain and vehicle dynamics), chassis domain (body and comfort), cockpit domain (infotainment and in-vehicle experience), autonomous driving domain (ADAS and highly automated driving) and body domain (connectivity), as shown in Figure 1. Each region correspondingly launches a corresponding domain controller, and finally connects to the main line or even hosts to the cloud through a Controller Area Network (CAN) /Local Interconnect Network (LIN) and other communication methods, so as to realize the interaction of vehicle information data [20,21,22]. The ECUs contained in the five domains are distinguished by different colors. In the domain-based architecture, ECUs belonging to the same domain are deployed in various positions of the vehicle body as required. At the same time, some lines between the corresponding domain controller and ECUs must cross the whole vehicle body, which is also the reason why the wiring harness length of the domain-based architecture is usually longer than that of the zonal architecture.

The power domain controller mainly controls the powertrain of the vehicle, optimizes the power performance of the vehicle, and ensures the power safety of the vehicle. The functions of the power domain controller include but are not limited to engine management, gearbox management, battery management, power distribution management, emission management, speed limit management, fuel saving and power saving management.

The chassis domain controller mainly controls the driving behavior and driving attitude of the vehicle. Its functions include, but are not limited to, braking system management, vehicle transmission system management, driving system management, steering system management, vehicle speed sensor management, body attitude sensor management, air suspension system management, and airbag system management.

The body domain controller mainly controls various body functions, including but not limited to headlights, rear lights, interior lights, door locks, windows, sunroofs, wipers, electric trunks, smart keys, air conditioners, antennas, and gateway communication.

The cockpit domain controller mainly controls various electronic information system functions in the intelligent cockpit of the vehicle. These functions include the central control system, vehicle infotainment system, head-up display, seat system, instrument system, rearview mirror system, driving behavior monitoring system, navigation system.

The automatic driving domain controller is responsible for realizing and controlling the automatic driving function of the car. It needs to have the ability to receive image information, process and judge image information, process and calculate data, navigate, and route planning, For the rapid judgment and decision-making ability of real-time situations, algorithms at the three levels of perception, decision-making, and control need to be processed, and the software and hardware requirements for domain controllers are the highest.

#### 2.2.2. Zonal IVN Architecture

In the zonal architecture, there are no longer traditional body domains, power domains, etc., and these are replaced by ‘zones’ in physical space, such as left front zone, right front zone, left rear zone, and right rear zone (Figure 2 shown). We use the same color as Figure 1 to mark the ECUs that belong to different zonal gateways to indicate that, in the zonal architecture, the ECUs are divided into different zones according to their physical location, while retaining the original ECUs deployment location in the domain-based architecture. Data interaction between the ECUs and central computers is realized through the zonal gateway. In the zonal architecture, the wiring harness structure becomes simpler, and the software environment becomes more flexible and scalable. Through over-the-air technology (OTA) updates, the new generation of car software can be easily upgraded to better support user-defined cars.

Audi, General Motors, Toyota, Jaguar, Land Rover, Volkswagen, and Volvo have announced that they will adopt the zonal architecture [23,24]. The reason why this architecture is adopted is to reduce the cost and weight of the wiring harness on the one hand, and to correspond to automatic driving on the other hand. The wiring harness is the third most expensive component in a mid-range car. The first is the engine and the second is the chassis. The most significant advantage of this architecture is that it can greatly reduce the wiring harness. Early research showed an approximate 30% reduction in weight and cost [25]. It can be further reduced if further optimized.

Considering that there is also a layered architecture approach, there are multiple wired communication technologies to suit various needs. Fast Ethernet is the dominant networking technology between the central computer and domain or zonal controllers. Currently, it is based on 100 Mb/s Ethernet, which will soon be upgraded to 1 Gb/s, and will be further supplemented by the new multi-gigabit technology defined by the IEEE [26,27]. For high security and reliability needs, redundancy can be added. There are still legacy bus systems such as Controller Area Network Flexible Data-Rate (CAN FD), LIN and FlexRay™. These technologies will remain in place for a period of time to ensure a smooth transition. In addition, various new standards are emerging for vehicle networking. On the one hand, there are various bus systems that can meet network requirements at a lower cost than conventional bus systems. On the other hand, there are also new popular schemes that achieve better performance/cost ratios. Compared with CAN FD, Controller Area Network Extra Long (CAN XL) has been upgraded in function and performance, has higher data rate, higher scalability, better security, and is compatible with traditional Ethernet based on switching topology (such as 100BASE-T1 and 1000BASE-T1), the channel of the Ethernet frame has a larger capacity. Although the CAN bus has many advantages, it does have its limitations. As the number of devices required to connect to a single CAN bus continues to increase, each new device significantly reduces performance due to its low data transfer rate of 5 Mb/s (CAN-FD) or 1 Mb/s (CAN). The application of Ethernet brings higher throughput (up to 10 Gbps or higher), and, compared with the maximum capacity of Data Field in CAN bus frame is only 8 bytes, a single frame of Ethernet can carry 44–1500 bytes of data. Additionally, Ethernet allows the aggregation of multiple CAN buses into one Ethernet link. This results in a smaller harness than CAN. This means lower installation and maintenance costs [28,29].

To realize the zonal architecture, a gigabit ethernet backbone network, time-sensitive network (TSN), adaptive automotive open system architecture (AUTOSAR) platform, high-performance computing (HPC), TSN switching domain controller, virtual ethernet switching, and IP/VLAN are all indispensable. The in-car application server (ICAS) system of Volkswagen MEB is provided by Continental, and the key Ethernet switch 88Q5050 is jointly developed by Elektrobit and Marvell [30]. At the same time, the NVIDIA DRIVE Pegasus self-driving car platform that NVIDIA claims can correspond to L5 is the first product to adopt 88Q5050. 88Q5050 can correspond to gigabit ethernet backbone network, TSN time-sensitive network, adaptive AUTOSAR platform, virtual ethernet switching, and internet protocol (IP)/virtual local area network (VLAN).

## 3. Comparison of Domain and Zonal IRN Architecture for Humanoid Robots

This section introduces the domain IRN architecture (DIRA) and the zonal IRN architecture (ZIRA) designed based on the types of sensors, detectors, processors, and actuators used by humanoid robots. In addition, the types and quantities of components included in the architectures proposed in this section refer to Taehyoung Kim’s paper [31]. In this paper, by collating medical and biological data, the sensor types suitable for humanoid robot sensor networks and the network parameters such as payload (only the data size is considered but not the entire frame size including the header file), bandwidth and data packet types are proposed.

### 3.1. Domain IRN Architecture

Figure 3 shows the DIRA designed in this paper for humanoid robots. The DIRA is divided into four domains: Head Domain (HD), Arm and Hand Domain (AHD), Leg and Foot Domain (LFD), and Skin Domain (SD). The four domains share a domain gateway to realize data interaction. In HD, the temperature and pressure sensors in the four parts of the forehead, nose, cheek, and lips, as well as the smell sensor and camera collect environmental information, and transmit it to the central processing unit (CPU) through the head domain controller (HDC). After the CPU analyzes and processes the data received from each domain controller (DC), it transmits control commands to the actuators located in HD, AHD, and LFD.

In addition, in AHD and LFD, according to the physical position of the robot’s limbs, the lower-level network is further divided, and the data transmission of the left arm, right arm, upper body, left leg, and right leg is processed respectively through five sub-domain controllers (SDC). The SD contains a large number of temperature and pressure sensors, which are used to collect the subtle environmental changes felt by the skin of humanoid robots. Therefore, the lower-level network connected to the SDC is also divided into three SDCs: the upper body, the arms, and the legs according to the physical location.

### 3.2. Zonal IRN Architecture

Figure 4 shows the ZIRA designed in this paper by reorganizing the DIRA based on the zonal architecture according to the IVN. The ZIRA is divided into six zones: head zone (HZ), left arm zone (LAZ), right arm zone (RAZ), torso zone (TZ), left leg zone (LLZ), and right leg zone (RLZ). Among them, LAZ, RAZ, TZ, LLZ, and RLZ are connected to the head zone gateway (HZG) of HZ through their own zone gateway (ZG), and all the links use 100 Mbps Ethernet.

The sensors, processors and actuators contained in the HZ are the same as DIRA. The LAZ and RAZ include the temperature and pressure sensor of arm and hand, the finger pressure sensor, and the actuators of the arm and hand on the left and right sides. The TZ contains the temperature and pressure sensors for chest and abdomen and back, and actuators for waist. The LLZ and RLZ contain the temperature sensor of leg and foot, the pressure sensor of calf, thigh, and foot, and the actuators of leg and foot on the left and right sides.

## 4. Wiring Harness Comparison of Domain and Zonal IRN Architecture

In this section, we show in detail the communication direction, data type, payload size, ethernet type, and link length of the network link of DIRA and ZIRA. Then, according to different levels of network components, the total link length and weight parameter of the wiring harness under the two architectures are calculated for comparison.

### 4.1. Data Transmission Link Parameter of Domain IRN Architecture

Table 1 shows the data types and average link lengths of temperature, pressure, smell sensors, and camera in DIRA’s HD which transmit environmental parameters to the HDC, and the links connect the HDC with the CPU and actuators. In this part, the sensor link is responsible for one-way uplink data transmission, while the actuator link carries one-way downlink data forwarding. Only the link of the CPU performs two-way data transmission at the same time.

Table 2 shows the link direction, data type, payload size, Ethernet bandwidth, and average link length of SSDC1 connected to the upper body in SD of DIRA; SSDC2 connected to the arm temperature and pressure sensor; and SSDC3 connected to the leg temperature and pressure sensor.

Table 3 integrates the actuator link specific parameters of the AHD and LFD, and this part of the link carries control data for one-way downlink transmission. Table 4 shows the link parameters between the DG and each domain.

### 4.2. Data Transmission Link Parameter of the Zonal IRN Architecture

This section shows the parameters of the links carried by the six zones in ZIRA. As the distribution of the components in the other five zones is different from that of DIRA except for the HZ, sensors and actuators are arranged according to the physical locations, so the average length of most links is shortened. For example, in the DIRA’s SD, the SDC located in the upper body of the humanoid robot carryies the data forwarded from the chest, back, arms, and legs, so it is inevitable to need a longer harness to connect the SDC with the sensors located at the legs and feet.

Table 5, Table 6, Table 7, Table 8, Table 9 and Table 10 show the communication direction, data type, payload size, Ethernet type, and link length of the sensor, processor, and actuator data transmission link of HZ, LAZ, RAZ, TZ, LLZ, and RLZ in turn. Except that the configuration of HZ is similar to HD in DIRA, other zones are responsible for the uplink and downlink data transmission of sensors and actuators connected to them according to the physical location of the components. Table 11 shows the link parameters of the 5 ZIRA zones connected to HZ.

### 4.3. Wiring Harness Weight and Length Comparison of DIRA and ZIRA

In this section, based on the average length of each link of the DIRA and the ZIRA, the difference in the total link length and link weight required by the two architectures is estimated when the humanoid robot is equipped with different numbers of sensors, processors, and actuators.

Among the existing humanoid robots developed for various fields, the average number of sensors used in each joint is less than 10, and the average number of actuators in each joint is five. LiDAR, camera, and other environmental parameter collection components are usually only equipped with one or two pieces. All data is handled by a single processor. Therefore, we calculated the link length and weight of the DIRA and the ZIRA based on the number of components mentioned above. In addition, on this basis, the required link length and weight are calculated when there are 5 times, 10 times, 20 times, 50 times, and 100 times the number of components in a current level humanoid which assumes 92 components as shown in Table 12. Among them, the number of smell sensor, camera, and CPU remains unchanged, and only the number of temperature sensors, pressure sensors, and actuators is adjusted.

The unit length weight of cables used in the calculation refers to the QSFP-40G-AOCxM cables. The QSFP-40G-AOCxM cable is QSFP+ active optical cables (AOC) for 40G Ethernet (40GbE) and InfiniBand QDR applications and compliant to the IEEE802.3ba (40GBASE-SR4). It supports bidirectional data transmission up to 4×10 Gbps, unit length weight is 0.1 kg (1 m). The total wiring harness length $L_{TD}$ of DIRA and $L_{TZ}$ of ZIRA can be calculated by Equations (1) and (2):(1)$$L_{TD}=\sum_{m=0}^{N_{S}-1}L_{DS_{m}}+\sum_{n=0}^{N_{A}-1}L_{DA_{n}}+\sum_{i=0}^{N_{DD}-1}L_{DD_{i}}+\sum_{j=0}^{N_{SD}-1}L_{SD_{j}}+2\times L_{DC}+L_{DT}+L_{DP},$$
where $N_{S}$ represents the total number of temperature and pressure sensors; $N_{A}$ represents the total number of actuators; $N_{DD}$ represents the total number of links connecting each domain controller and backbone domain gateway; $N_{SD}$ is the total number of lines connecting each domain controller and sub-domain controller; $L_{DS_{m}}$ is the length of the link between the $m^{th}$ sensor and the sub-domain controllers; $L_{DA_{n}}$ represents the length of link between the $n^{th}$ actuator and the sub-domain controllers; $L_{DD_{i}}$ represents the length of the connecting link between the $i^{th}$ domain controller and domain gateway; $L_{SD_{j}}$ represents the length of the link between the $j^{th}$ sub-domain controller and the domain controllers; $L_{DC}$ represents the length of the link between the camera and the head domain controller; $L_{DT}$ represents the link length between the smell sensor and the head domain controller; $L_{DP}$ represents the link length between the CPU and the head domain controller. The total length of the entire DIRA architecture wiring harness is obtained by summing the lengths of the individual parts.
(2)$$L_{TZ}=\sum_{m=0}^{N_{S}-1}L_{ZS_{m}}+\sum_{n=0}^{N_{A}-1}L_{ZA_{n}}+\sum_{i=0}^{N_{ZZ}-1}L_{ZZ_{i}}+2\times L_{ZC}+L_{ZT}+L_{ZP},$$
where $L_{ZS_{m}}$ is the length of the link between the $m^{th}$ sensor and the zone gateways; $L_{ZA_{n}}$ represents the length of link between the $n^{th}$ actuator and the zone gateways; $L_{ZZ_{i}}$ is the length of the connecting link between head zone gateway and the $i^{th}$ zone gateway; $L_{ZC}$ represents the length of the link between the camera and the head zone gateway; $L_{ZT}$ is the link length between the smell sensor and the head zone gateway; $L_{ZP}$ represents the link length between the CPU and the head zone gateway. The total length of the entire ZIRA architecture wiring harness is obtained by summing the lengths of the individual parts.

The total weight of the wiring harness of DIRA and ZIRA architecture can be calculated by Equations (3) and (4).
(3)$$W_{TD}=W_{F}\times L_{TD}$$
(4)$$W_{TZ}=W_{F}\times L_{TZ}$$
where $W_{F}$ represents the weight of the wiring harness per unit length. In this paper, the value 0.1 kg/m mentioned above is used to calculate the total weight of the wiring harness for DIRA and ZIRA: $W_{TD}$ is the total wiring harness weight of DIRA, and $W_{TZ}$ is the total wiring harness weight of ZIRA.

Table 12 shows the total number of components carried by the DIRA and the ZIRA, and the calculation results of the total wiring harness length and weight between the two architectures when equipped with different numbers of temperature and pressure sensors. Among them, the change in the number of components is only given to the temperature and pressure sensors, and the other components remain unchanged: one smell sensor, two cameras, and one CPU. The 1-time quantity represents the number of components carried by the currently released humanoid robot, with an average of two sensors and two actuators per joint. Based on this, we calculate the difference in the length and weight of the harness when the number of components increases to 5 times, 10 times, 20 times, 50 times, and 100 times the current level.

The results in Figure 5 show that when each joint is equipped with two sensors and actuators (i.e., the total number of components is 92), the total length and total weight of the ZIRA wiring harness are 16.71% less than the DIRA. As the number of components increases, the difference in total length and total weight between the two architectures gradually increases. When the total number of components reaches 8804, the length and weight of ZIRA are 17.50% less than that of DIRA. Numerically, the total length of the ZIRA’s wiring harness is 704.95 m shorter than that of the DIRA, while the weight difference is 70.5 kg.

## 5. Conclusions

In this paper, by referring to the development trend of the electronic components highly integrated IVN field, we found that compared with domain architecture, zonal architecture has several advantages in network expansion, wiring complexity, vehicle body weight control, and management and maintenance. We applied it to the IRN architecture design and compared it with the domain architecture in several aspects.

We introduced the differences in the structure of sensor networks for humanoid robots under the proposed ZIRA and the previous DIRA concept. Through careful analysis of numerical simulations, the difference in the length and weight of the wiring harness of the two network architectures was compared under the different quantities of sensors and actuators. The results show that at the level of sensor integration of the humanoid robots today, there is a significant improvement of more than 16% in the length and weight of the wiring harnesses between the two architectures. In the future, as humanoid robots are equipped with more sensors and actuators, these improvements may exponentially increase as shown in Figure 5.

In future research, we will further change the network parameters such as number of ZIRA components, sensor network topology, and payload data, and determine the performance difference between the ZIRA and the DIRA in the data transmission process through the network simulation under different structures.

## Figures and Tables

**Figure 1 sensors-23-02627-f001:**
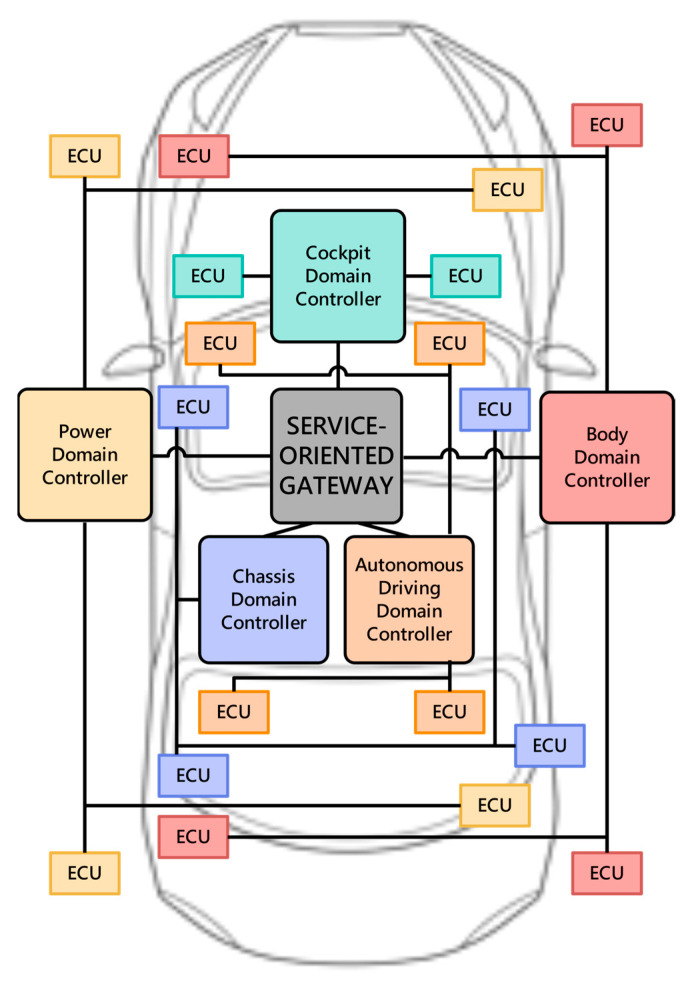
Body structure of domain IVN architecture. The network is divided into five areas by function: power domain, chassis domain, cockpit domain, autonomous driving domain and body domain.

**Figure 2 sensors-23-02627-f002:**
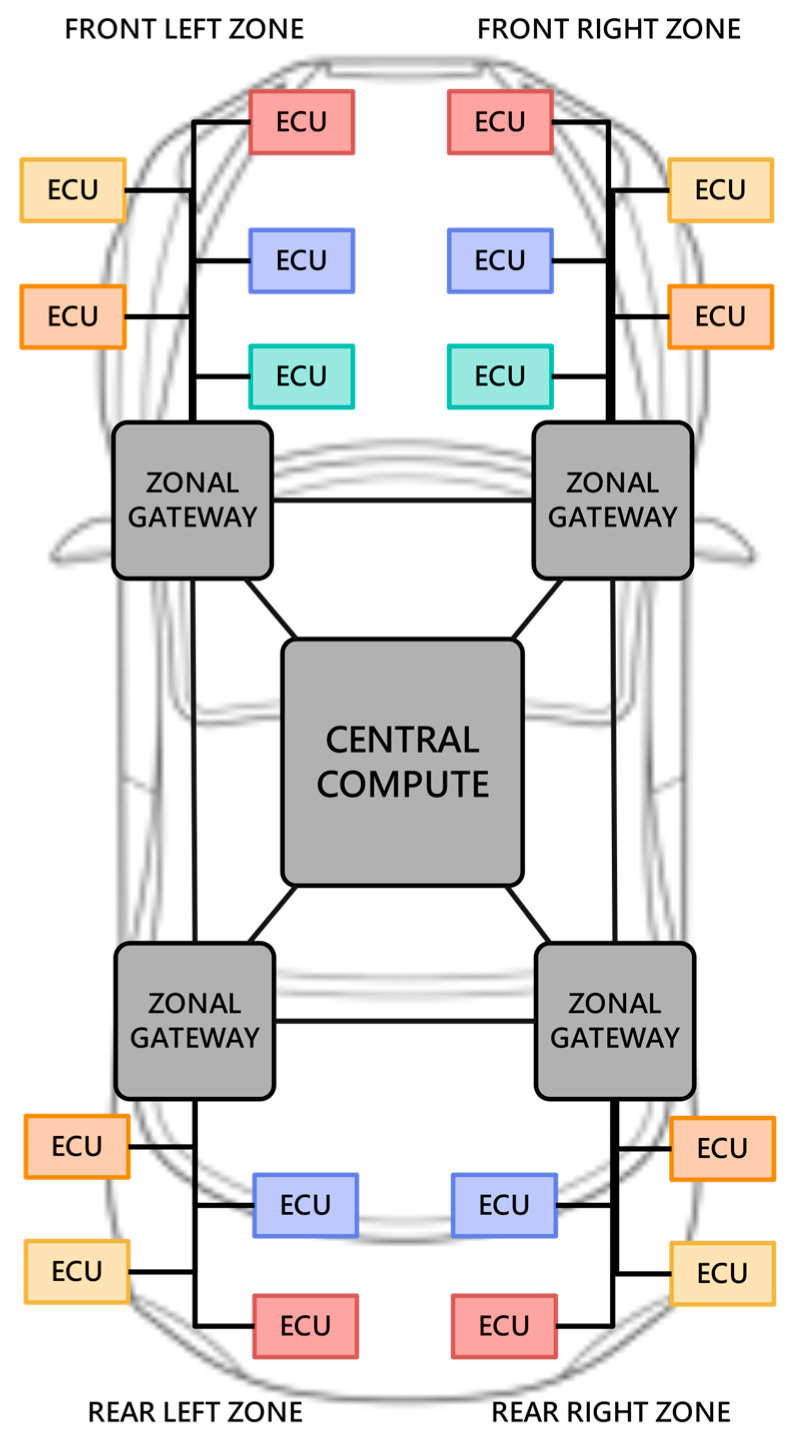
Body structure of zonal IVN architecture. The network is replaced in physical space, includes left front zone, right front zone, left rear zone, and right rear zone.

**Figure 3 sensors-23-02627-f003:**
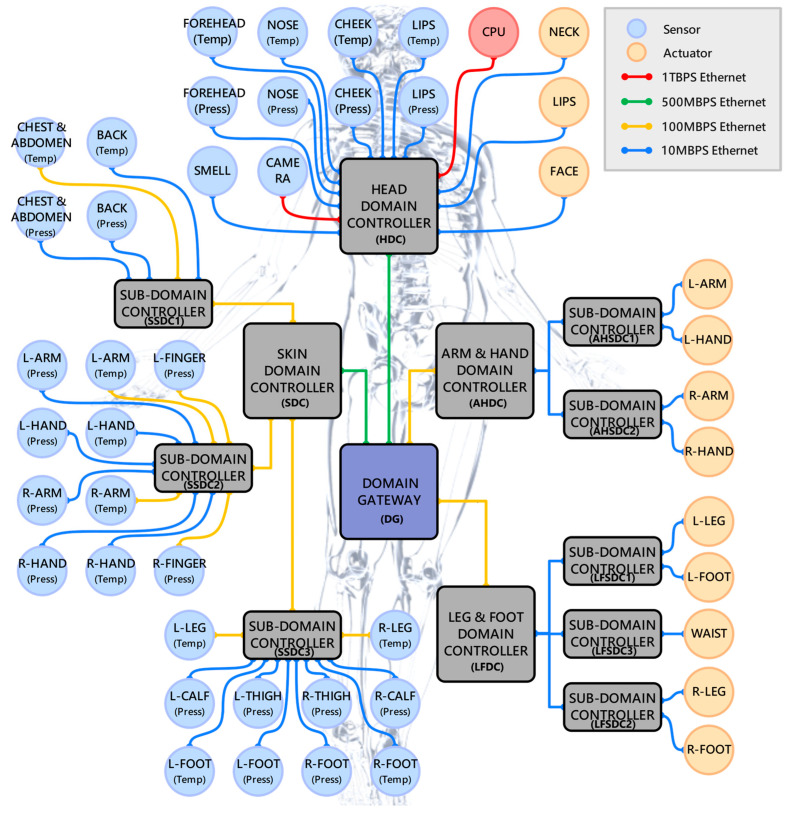
Body structure of domain IRN architecture. The network is divided into Head Domain, Arm and Hand Domain, Leg and Foot Domain, and Skin Domain.

**Figure 4 sensors-23-02627-f004:**
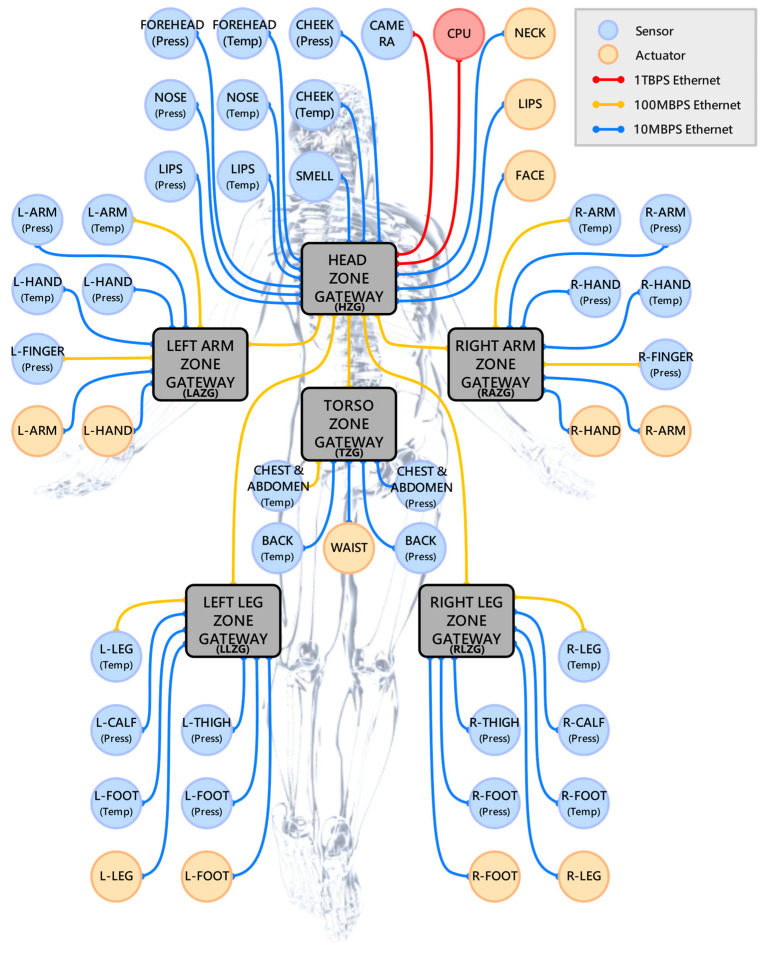
Body structure of zonal IRN architecture. The network is physically divided into six zones: head zone, left arm zone, right arm zone, torso zone, left leg zone, and right leg zone.

**Figure 5 sensors-23-02627-f005:**
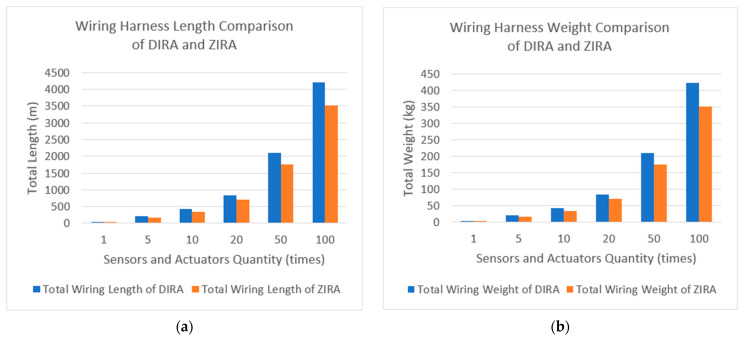
The total length and total weight of the DIRA and ZIRA under different component quantities: (**a**) describes the difference in the total wiring length when DIRA and ZIRA are equipped with different numbers of temperature and pressure sensors. (**b**) describes the difference in the total wiring weight when DIRA and ZIRA are equipped with different numbers of temperature and pressure sensors.

**Table 1 sensors-23-02627-t001:** Data transmission link parameters of the DIRA Head Domain.

Transmission Path	Traffic Type	Payload Size (Byte)	Ethernet Bandwidth	Average Length (m)
Forehead (Temp) → HDC	Sensor	232	10 Mbps	0.32
Forehead (Press) → HDC	Sensor	14	10 Mbps	0.32
Nose (Temp) → HDC	Sensor	130	10 Mbps	0.22
Nose (Press) → HDC	Sensor	30	10 Mbps	0.22
Cheek (Temp) → HDC	Sensor	31.96 k	10 Mbps	0.20
Cheek (Press) → HDC	Sensor	9.86 k	10 Mbps	0.20
Lips (Temp) → HDC	Sensor	92	10 Mbps	0.18
Lips (Press) → HDC	Sensor	20	10 Mbps	0.18
Smell → HDC	Sensor	1000	10 Mbps	0.22
Camera → HDC	Sensor	3.46 G	1 Tbps	0.28
HDC ↔ CPU	Sensor and Control	N/A	1 Tbps	0.30
HDC → Neck	Control	10	10 Mbps	0.07
HDC → Lips	Control	6	10 Mbps	0.18
HDC → Face	Control	80	10 Mbps	0.20

**Table 2 sensors-23-02627-t002:** Data transmission link parameters of the DIRA Skin Domain.

Transmission Path	Traffic Type	Payload Size (Byte)	Ethernet Bandwidth	Average Length (m)
Chest and Abdomen (Temp) → SSDC1	Sensor	52.99 k	100 Mbps	0.20
Chest and Abdomen (Press) → SSDC1	Sensor	492	10 Mbps	0.20
Back (Temp) → SSDC1	Sensor	33.02 k	10 Mbps	0.20
Back (Press) → SSDC1	Sensor	2.71 k	10 Mbps	0.20
SSDC1 → SDC	Sensor	N/A	100 Mbps	0.15
L-Arm (Temp) → SSDC2	Sensor	38.4 k	100 Mbps	0.47
L-Arm (Press) → SSDC2	Sensor	2.54 k	10 Mbps	0.47
L-Finger (Press) → SSDC2	Sensor	9.60 k	100 Mbps	0.55
L-Hand (Temp) → SSDC2	Sensor	9.60 k	10 Mbps	0.51
L-Hand (Press) → SSDC2	Sensor	600	10 Mbps	0.51
R-Arm (Temp) → SSDC2	Sensor	38.4 k	100 Mbps	0.47
R-Arm (Press) → SSDC2	Sensor	2.54 k	10 Mbps	0.47
R-Finger (Press) → SSDC2	Sensor	9.60 k	100 Mbps	0.55
R-Hand (Temp) → SSDC2	Sensor	9.60 k	10 Mbps	0.51
R-Hand (Press) → SSDC2	Sensor	600	10 Mbps	0.51
SSDC2 → SDC	Sensor	N/A	100 Mbps	0.20
L-Leg (Temp) → SSDC3	Sensor	49.53 k	100 Mbps	0.65
L-Calf (Press) → SSDC3	Sensor	1.62 k	10 Mbps	0.90
L-Thigh (Press) → SSDC3	Sensor	2.94 k	10 Mbps	0.45
L-Foot (Temp) → SSDC3	Sensor	11.90 k	10 Mbps	1.15
L-Foot (Press) → SSDC3	Sensor	372	10 Mbps	1.15
R-Leg (Temp) → SSDC3	Sensor	49.53 k	100 Mbps	0.65
R-Calf (Press) → SSDC3	Sensor	1.62 k	10 Mbps	0.90
R-Thigh (Press) → SSDC3	Sensor	2.94 k	10 Mbps	0.45
R-Foot (Temp) → SSDC3	Sensor	11.90 k	10 Mbps	1.15
R-Foot (Press) → SSDC3	Sensor	372	10 Mbps	1.15
SSDC3 → SDC	Sensor	N/A	100 Mbps	0.30

**Table 3 sensors-23-02627-t003:** Data transmission link parameters of the DIRA Arm and Hand Domain, Leg and Foot Domain.

Transmission Path	Traffic Type	Payload Size (Byte)	Ethernet Bandwidth	Average Length (m)
AHSDC1 → L-Arm	Control	12	10 Mbps	0.27
AHSDC1 → L-Hand	Control	5	10 Mbps	0.31
AHSDC2 → R-Arm	Control	12	10 Mbps	0.27
AHSDC2 → R-Hand	Control	5	10 Mbps	0.31
AHDC → AHSDC1	Control	N/A	10 Mbps	0.20
AHDC → AHSDC2	Control	N/A	10 Mbps	0.20
LFSDC1 → L-Leg	Control	19	10 Mbps	0.50
LFSDC1 → L-Foot	Control	6	10 Mbps	1.00
LFSDC2 → R-Leg	Control	19	10 Mbps	0.50
LFSDC2 → R-Foot	Control	6	10 Mbps	1.00
LFSDC3 →Waist	Control	55	10 Mbps	0.20
LFDC → LFSDC1	Control	N/A	10 Mbps	0.15
LFDC → LFSDC2	Control	N/A	10 Mbps	0.15
LFDC → LFSDC3	Control	N/A	10 Mbps	0.20

**Table 4 sensors-23-02627-t004:** Data transmission link parameter of the link connects the DIRA Domain Gateway and each Domain Controllers.

Transmission Path	Traffic Type	Ethernet Bandwidth	Average Length (m)
Domain Gateway ↔ HDC	Sensor and Control	500 Mbps	0.27
SDC → Domain Gateway	Sensor	500 Mbps	0.31
Domain Gateway ↔ AHDC	Control	100 Mbps	0.27
Domain Gateway ↔ LFDC	Control	100 Mbps	0.31

**Table 5 sensors-23-02627-t005:** Data transmission link parameters of the ZIRA Head Zone.

Transmission Path	Traffic Type	Payload Size (Byte)	Ethernet Bandwidth	Average Length (m)
Forehead (Temp) → HZG	Sensor	232	10 Mbps	0.32
Forehead (Press) → HZG	Sensor	14	10 Mbps	0.32
Nose (Temp) → HZG	Sensor	130	10 Mbps	0.22
Nose (Press) → HZG	Sensor	30	10 Mbps	0.22
Cheek (Temp) → HZG	Sensor	31.96 k	10 Mbps	0.20
Cheek (Press) → HZG	Sensor	9.86 k	10 Mbps	0.20
Lips (Temp) → HZG	Sensor	92	10 Mbps	0.18
Lips (Press) → HZG	Sensor	20	10 Mbps	0.18
Smell → HZG	Sensor	1000	10 Mbps	0.22
Camera → HZG	Sensor	3.46 G	1 Tbps	0.28
HZG ↔ CPU	Sensor and Control	N/A	1 Tbps	0.30
HZG → Neck	Control	10	10 Mbps	0.07
HZG → Lips	Control	6	10 Mbps	0.18
HZG → Face	Control	80	10 Mbps	0.18

**Table 6 sensors-23-02627-t006:** Data transmission link parameters of the ZIRA Left Arm and Hand Zone.

Transmission Path	Traffic Type	Payload Size (Byte)	Ethernet Bandwidth	Average Length (m)
L-Arm (Temp) → LAZG	Sensor	38.4 k	100 Mbps	0.27
L-Arm (Press) → LAZG	Sensor	2.54 k	10 Mbps	0.27
L-Finger (Press) → LAZG	Sensor	9.60 k	100 Mbps	0.35
L-Hand (Temp) → LAZG	Sensor	9.60 k	10 Mbps	0.31
L-Hand (Press) → LAZG	Sensor	600	10 Mbps	0.31
LAZG → L-Arm	Control	12	10 Mbps	0.27
LAZG → L-Hand	Control	5	10 Mbps	0.31

**Table 7 sensors-23-02627-t007:** Data transmission link parameters of the ZIRA Right Arm and Hand Zone.

Transmission Path	Traffic Type	Payload Size (Byte)	Ethernet Bandwidth	Average Length (m)
R-Arm (Temp) → RAZG	Sensor	38.4 k	100 Mbps	0.27
R-Arm (Press) → RAZG	Sensor	2.54 k	10 Mbps	0.27
R-Finger (Press) → RAZG	Sensor	9.60 k	100 Mbps	0.35
R-Hand (Temp) → RAZG	Sensor	9.60 k	10 Mbps	0.31
R-Hand (Press) → RAZG	Sensor	600	10 Mbps	0.31
RAZG → R-Arm	Control	12	10 Mbps	0.27
RAZG → R-Hand	Control	5	10 Mbps	0.31

**Table 8 sensors-23-02627-t008:** Data transmission link parameters of the ZIRA Torso Zone.

Transmission Path	Traffic Type	Payload Size (Byte)	Ethernet Bandwidth	Average Length (m)
Chest and Abdomen (Temp) → TZG	Sensor	52.99 k	100 Mbps	0.20
Chest and Abdomen (Press) → TZG	Sensor	492	10 Mbps	0.20
Back (Temp) → TZG	Sensor	33.02 k	10 Mbps	0.20
Back (Press) → TZG	Sensor	2.71 k	10 Mbps	0.20
TZG →Waist	Control	55	10 Mbps	0.20

**Table 9 sensors-23-02627-t009:** Data transmission link parameters of the ZIRA Left Leg and Foot Zone.

Transmission Path	Traffic Type	Payload Size (Byte)	Ethernet Bandwidth	Average Length (m)
L-Leg (Temp) → LLZG	Sensor	49.53 k	100 Mbps	0.50
L-Calf (Press) → LLZG	Sensor	1.62 k	10 Mbps	0.75
L-Thigh (Press) → LLZG	Sensor	2.94 k	10 Mbps	0.30
L-Foot (Temp) → LLZG	Sensor	11.90 k	10 Mbps	1.00
L-Foot (Press) → LLZG	Sensor	372	10 Mbps	1.00
LLZG → L-Leg	Control	19	10 Mbps	0.50
LLZG → L-Foot	Control	6	10 Mbps	1.00

**Table 10 sensors-23-02627-t010:** Data transmission link parameters of the ZIRA Right Leg and Foot Zone.

Transmission Path	Traffic Type	Payload Size (Byte)	Ethernet Bandwidth	Average Length (m)
R-Leg (Temp) → RLZG	Sensor	49.53 k	100 Mbps	0.50
R-Calf (Press) → RLZG	Sensor	1.62 k	10 Mbps	0.75
R-Thigh (Press) → RLZG	Sensor	2.94 k	10 Mbps	0.30
R-Foot (Temp) → RLZG	Sensor	11.90 k	10 Mbps	1.00
R-Foot (Press) → RLZG	Sensor	372	10 Mbps	1.00
RLZG → R-Leg	Control	19	10 Mbps	0.50
RLZG → R-Foot	Control	6	10 Mbps	1.00

**Table 11 sensors-23-02627-t011:** Data transmission link parameters of the link connecting the Head Zone Gateway and each Zone Gateways.

Transmission Path	Traffic Type	Ethernet Bandwidth	Average Length (m)
Head Zone Gateway ↔ Torso Zone Gateway	Sensor and Control	100 Mbps	0.10
Head Zone Gateway ↔ Left Arm Zone Gateway	Sensor and Control	100 Mbps	0.36
Head Zone Gateway ↔ Right Arm Zone Gateway	Sensor and Control	100 Mbps	0.36
Head Zone Gateway ↔ Left Leg Zone Gateway	Sensor and Control	100 Mbps	0.47
Head Zone Gateway ↔ Right Leg Zone Gateway	Sensor and Control	100 Mbps	0.47

**Table 12 sensors-23-02627-t012:** The Total Length and Total Weight Comparison of the DIRA and ZIRA Architecture Wiring Harness under different quantities of sensors and actuators.

Sensors and Actuator Quantity	Total Component Number	DIRA Total Length (m)	ZIRA Total Length (m)	DIRA Total Weight (kg)	ZIRA Total Weight (kg)
1 time	92	45.65	37.66	4.56	3.76
5 times	444	214.21	178.06	21.42	17.80
10 times	884	424.91	353.56	42.49	35.35
20 times	1764	846.31	704.56	84.63	70.45
50 times	4404	2110.51	1757.56	211.05	175.75
100 times	8804	4217.51	3512.56	421.75	351.25

## Data Availability

Not applicable.

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
