# Peer review of "Physical Length and Weight Reduction of Humanoid In-Robot Network with Zonal Architecture"

_sensors, 2023, doi:10.3390/s23052627_

Round 1

Reviewer 1 Report

 In this paper, the process of design sensor networks suitable for humanoid robots to designing an in-robot network (IRN) that can support a large sensor network for reliable data exchange. The paper is well written and addresses a problem of interest. The results are analyzed well and compared with the existing methods. I don't find any major issues or flaws in the article. I recommend that the paper can be accepted for publication in its current form.

Author Response

Dear Editors and Reviewers:

 Thank you for your letter and for the reviewers’ comments concerning our manuscript entitled “Physical Length and Weight Reduction of Humanoid In-Robot Network with Zonal Architecture” (ID: sensors-2202642). Those comments are all valuable and very helpful for revising and improving our paper, as well as the important guiding significance to our researches. We have studied comments carefully and have made correction which we hope meet with approval.

Reviewer 2 Report

This paper introduced the differences between the zonal IRN (in-robot network) architecture ZIRA and the domain-based IRN architecture DIRA in terms of network structure, hardware quantity, and regional division for a humanoid robot. The author showed the differences in the total link length of the network and the weight parameter of the wiring harness between the two architectures. The results show that the total length and total weight of the ZIRA wiring harness are 16.71% less than DIRA. The reviewer considers that this result is very useful for a humanoid robot to reduce weight. As stated in the conclusion, a detailed explanation for the performance difference between ZIRA and DIRA through the network simulation is needed.

The reviewer points out several mistakes shown below.

1.    The sentences in 2.2 and 2.2.1 are redundant.

2.    The text is interspersed before and after Table 1.

3.    There are a few writing errors in DIRA and HIRA in Figure 5.

Author Response

Dear Editors and Reviewers:

 Thank you for your letter and for the reviewers’ comments concerning our manuscript entitled “Physical Length and Weight Reduction of Humanoid In-Robot Network with Zonal Architecture” (ID: sensors-2202642). Those comments are all valuable and very helpful for revising and improving our paper, as well as the important guiding significance to our researches. We have studied comments carefully and have made correction which we hope meet with approval. The main corrections in the paper and the responds to the reviewer’s comments are as flowing:

Response 1:
The redundant sentences in 2.2 and 2.2.1 have been deleted.

Response 2:
Table 1 has been moved to the correct location.

Response 3:
The wrting errors in Figure 5 have been corrected, HIRA has been corrected to DIRA

Additionally, we redraw figure 1 and added its description on lines 139-145.
We have revised the text of the article on lines 190-192.
We have redrawn figure 2 and added its description on lines 178-183.
We have added the full text to some abbreviations in the paper, respectively in lines 52-53, 137-138, 204, 210, and 231.
We added some description on lines 213-221 to describe the difference between ethernet and CAN bus used for the In-Vehicle Networks.

Special thanks to you for your good comments.

Reviewer 3 Report

This paper compared two architectures of humanoid in robot networks and proposed a possible architecture solution. This results shows that this solution reduces the cost of the robot by using less electrical components, shorter wires, and lighter weight. This architecture has been demonstrated to be effective and will be adopted by some famous auto manufacturers. The novelty of this paper is not too impressive. Some other comments are listed below:  

1. The contents of 2.2 and the first paragraph of 2.2.1 are exactly the same.

2. In figure 1, the network is hard to read and figure out which part is power domain, chassis domain, etc., which are illustrated in the description. 3. The paper is written with some duplicate sentences, for example, in line 186-188, 'the third expensive component is wiring harness' is repeated twice. 4. In figure 2, what does the color background mean? They are all ECU, what is the difference? Also, the grey wires don't make sense to me. 5. There are abbreviations that are not defined in the paper, for example, CAN/LIN, FD, etc. Please check them. 6. In line 208, the author says 'the ethernet frame has a larger capacity', what is the reason? 7. The result in Line 389-393 is not convincible, why compare under the condition of 100 times sensors quantity? With the data analysis improvement, the amount of sensors is becoming less and less, the sensor amount of 100 times and total weight of 421.75 kg and 351.25kg is out of reality.

Author Response

Dear Editors and Reviewers:

 Thank you for your letter and for the reviewers’ comments concerning our manuscript entitled “Physical Length and Weight Reduction of Humanoid In-Robot Network with Zonal Architecture” (ID: sensors-2202642). Those comments are all valuable and very helpful for revising and improving our paper, as well as the important guiding significance to our researches. We have studied comments carefully and have made correction which we hope meet with approval. The main corrections in the paper and the responds to the reviewer’s comments are as flowing:

Response 1:
The redundant sentences in 2.2 and 2.2.1 have been deleted.

Response 2:
According to your comments, we have redrawn figure 1. We choose to use different colors to indicate the ECUs under different domains, and adjusted the position of the ECUs in the vehicle body in the picture. In addition, the names of each domain are clearly marked in the picture. We also added 139-145 lines to further describe figure 1.

Response 3:
We have made corrections to lines 190-192 of the paper based on your comments.

Response 4:
We redraw figure 2 and keep the colors used in figure 1 to distinguish different ECUs. Our purpose is to indicate that in the zonal architecture, different zones are divided according to the physical location of the ECUs without changing the position of the original ECUs inside the vehicle body. The intention is to indicate that ECUs that are originally located in similar locations and belong to different domains are divided into the same zone. In addition, we added lines 178-183 to further describe figure 2.

Response 5:
Based on your comments, we double-checked our paper and added full text for abbreviations that were not defined. The changes are in lines 52-53, 137-138, 204, 210 and 231.

Response 6:
We added a further description in lines 213-221 of the paper, explaining why the ethernet frame has a greater capacity by comparing the ethernet and CAN bus used in the vehicle's internal network.

Response 7:
Thanks for your valuable comment. We choose to compare the IRN wiring harness with 100 times the number of sensors because in the expectation of the future humanoid robot sensor network proposed in references 6 and 35 of this paper, in order to achieve a perception ability similar to human beings, it can be considered to equip the robot with tens of thousands of sensors. Of course, so far, the existing sensors are far from being able to realize this idea in terms of volume and weight. Therefore, in this paper, we increase the order of magnitude of sensors to 100 times that of existing robots(this is much smaller than the sensor network scale proposed in the references), and intend to compare the differences that deploying so many sensors will bring to wiring harnesses under different network architectures under this quantities. We believe that even if the current technology cannot realize the integration of such a large number of sensors, the purpose of this paper is to look forward to and expand the research in the field of future humanoid robots. Our results show that there is a difference of more than 16% in the weight and length of the wiring harness of the two network architectures, while the sensor quantity under the existing IRN or under 100 times of it. The total weight of 421.75 kg and 351.25 kg is also calculated under the existing hardware level conditions. We believe that in the future, with the improvement of the hardware level, the total weight of the wiring harness brought by such a huge sensor network will also be greatly reduced to a realistic range.

Special thanks again to you for your good comments.

Round 2

Reviewer 3 Report

This version has addressed all my comments of last version, I would recommend it qualified to publish.